# Time to incidence of tuberculosis and its predictors among adult HIV/AIDS patients who initiated ART by the Universal Test and Treat approach in Silte Zone, Ethiopia, 2023

Abdulbasit Sherfa[1]*, Kemal Lemnuro[2], Mohammed Muze[3], Abdulkerim Badegba Isa[1], Abdulmejid Mustefa Shemsu[1], Musa Jemal[1], Abas Ali Hassen[4], Dawit Tafesse Darsema[4], Belete Birhan[5], Wolyu Korma[1]

1 Department of Public Health, College of Medicine and Health Sciences, Werabe University, Werabe, Ethiopia, 2 Department of Medicine, College of Medicine and Health Sciences, Werabe University, Werabe, Ethiopia, 3 Department of Nursing, College of Medicine and Health Sciences, Werabe University, Werabe, Central Ethiopia Region, Ethiopia, 4 Department of Anesthesia, College of Medicine and Health Sciences, Werabe University, Werabe, Central Ethiopia Region, Ethiopia, 5 Department of Psychiatry, College of Medicine and Health Sciences, Wolayita Sodo University, Wolayita Sodo, Ethiopia

* abdusherfa3@gmail.com

## Abstract

Tuberculosis (TB) remains a leading cause of morbidity and mortality among people living with HIV. TB/HIV co-infection continues to challenge global TB control efforts. This study aimed to estimate the incidence and identify predictors of TB among adult HIV-infected patients who initiated antiretroviral therapy (ART) under the Universal Test and Treat (UTT) approach in Silte Zone, Ethiopia. An institution-based retrospective cohort study was conducted among 404 adult HIV patients enrolled in ART. Participants were selected using simple random sampling. Data were extracted using a structured checklist via Kobo Toolbox and analyzed using STATA version 14. Cox proportional hazards regression models were applied to identify predictors of TB. Statistical significance was declared at p < 0.05 with 95% confidence intervals. The proportional hazards assumption was assessed using statistical tests and graphical methods. The median age was 36 years, with near-equal sex distribution. Most participants initiated ART at WHO clinical stages I–II. Approximately 70% had good adherence, and over 80% disclosed their HIV status. The predominant regimen was TDF-3TC-EFV. The overall TB incidence density rate was 5.33 per 1000 person-months (95% CI: 3.68–7.77). The incidence of new TB was 4.0 per 1000 person-months (95% CI: 2.60–6.13), while reinfection was 1.3 per 1000 person-months (95% CI: 0.63–2.70). TB-free survival probabilities at 6, 12, and 18 months were 0.99, 0.93, and 0.90, respectively. Male sex (AHR: 5.05), non-disclosure of HIV status (AHR: 6.29), underweight status (AHR: 3.07), CD4 count <200 cells/μL (AHR: 5.63), and poor ART adherence (AHR: 7.05) were significant predictors. Although TB incidence declined under the UTT approach, risk remained elevated

**Data availability statement:** All relevant materials and data are within the paper and its Supporting Information files, and if any additional data or clarification is needed, it will be available from the corresponding author upon reasonable request.

**Funding:** The author(s) declare that financial support was received for the research of this article. Financial support was obtained from Werabe University (grant no: Ref/NOWRU/RPD/0001/2015EC). The funder had no involvement in the study's design, data collection, analysis, and interpretation, or manuscript preparation.

**Competing interests:** The authors have declared that no competing interests exist.

during the first year of ART. Targeted interventions promoting early diagnosis, adherence support, nutritional care, and safe disclosure are essential to reduce TB burden among people living with HIV.

## Introduction

Tuberculosis (TB) is the leading opportunistic infection and a major cause of death among people living with Human Immunodeficiency Virus/Acquired Immunodeficiency Syndrome (HIV/AIDS) [1]. Individuals with HIV are 20–30 times more likely to develop active TB, which accounts for about one-third of HIV-related deaths [2]. In 2022, 8% of all TB cases occurred in people with HIV [3]. TB hastens the progression of HIV, while HIV increases TB severity, raises the likelihood of recurrence and [4–6]; and also increases the chance of acquiring resistant TB [7]. Consequently, early diagnosis and prompt initiation of HIV treatment through the Universal Test and Treat (UTT) strategy are critical components of effective global TB control efforts [8].

Individuals with recurrent TB have a higher risk of death than those with first-time TB episodes, partly due to advanced immunosuppression and complications from repeated lung damage [9]. Recurrent TB also carries an increased likelihood of multidrug-resistant TB (MDR-TB), especially in cases where initial treatment was incomplete or poorly adhered to [10]. Furthermore, each episode can result in cumulative pulmonary impairment, reduced lung capacity, and diminished quality of life (QoL) [11]. At the public health level, recurrent TB sustains community transmission and complicates TB elimination efforts [12]. Among HIV-positive patients, recurrent TB can accelerate HIV disease progression through chronic immune activation and systemic inflammation, even in those receiving ART [13].

Tuberculosis (TB) remains a critical public health challenge among individuals living with HIV, significantly impeding the attainment of global TB prevention targets [14,15]. AIDS-related illnesses, notably TB and bacterial infections, are most common cause of hospital admission among adults with HIV across diverse geographic regions and are the leading cause of in-hospital mortality [16,17]. Evidence revealed that low adherence to antiretroviral treatment (ART) are correlated with significantly higher rates of AIDS-defining Opportunistic Infections (OIs), thereby compromising treatment effectiveness and imposing substantial economic burdens on health systems [18]. Moreover, post-mortem studies from Sub-Saharan Africa reveal a substantial prevalence of undiagnosed disseminated TB among HIV/AIDS patients, suggesting that TB-related mortality in this population is likely considerably underestimated [19,20].

In Ethiopia, pooled estimates from multiple studies indicate that the prevalence of TB/HIV co-infection is approximately 25.59% [21], markedly higher than the national average of 9% [22]. TB is the most frequently detected OI in both pre-ART and on-ART patients [23], and it is also the leading recurrent OIs, comprising 17.5% of cases among HIV patients [24]. Evidence suggests that several factors contribute to new or recurrent TB disease among people living with HIV/AIDS (PLHIV/AIDS), including Cell Differentiation 4 (CD4)lymphocyte count [25–28], Body Mass Index (BMI) [27,29], behavioral factors such as smoking and alcohol use, as well as sputum

smear test results [30–32], and poor adherence to ART and advanced HIV disease [33,34]. These variables have been commonly identified as predictors of new and recurrent TB. infection. To improve clinical outcomes, increase access to antiretroviral (ARV) drugs for treating and preventing HIV, and ultimately achieve the "goal of ending the HIV epidemic as a major public health threat by 2030," the World Health Organization (WHO) released a recommendation in 2016 urging the provision of UTT services for people living with HIV (PLHIV), irrespective of their CD4 cell count or WHO clinical stage [35]. The UTT initiative, the immediate or rapid initiation of ART within 1 week of HIV diagnosis, was introduced in Ethiopia in August 2016 [36].

Millions of people still get TB every year despite WHO efforts to reduce the incidence of TB diseases through the implementation of the Directly Observed Treatment strategy. The problem is that OIs linked to HIV can cause recurrence in patients who have already received treatment and been cured [24,37,38]. Tuberculosis is the leading cause of morbidity and mortality among people living with HIV (PLHIV) [39,40]. Since the beginning of the UTT program, PLHIV have been treated with a variety of intervention modalities to reduce the risk of TB/HIV co-infection as well as TB-related morbidity and mortality. The burden and factors that contribute to tuberculosis occurrence and recurrence have an impact on the health system and the population, making it difficult to meet the 2030 Sustainable Development Goals (SDGs). Evidence on the magnitude of TB in Ethiopia among PLHIV following UTT treatment is scarce. The purpose of this study is to assess the incidence and risk factors for tuberculosis among adult HIV patients in the Silte Zone who began receiving ART following the implementation of the UTT program.

## Methods

### Ethics statement

Ethical clearance for this study was obtained from the Institutional Ethical Review Board (IRB) of Werabe University prior to data collection (Reference: WRU/CMHS/09/2022; Protocol: GOV/WRU/CMHS/PH/2022–1415; dated May 2, 2022). The clearance letter was subsequently submitted to the Zonal Health Bureau, Silte Zone Health Facilities, and other relevant authorities for official permission to conduct the research. Formal approval was granted by the respective health facilities. To ensure compliance with ethical standards, confidentiality and anonymity of patient information were strictly maintained throughout the study. Data were extracted from medical records without any direct face to face patient contact, and no personally identifiable information was disclosed or recorded. For patients, the study's purpose and data protection measures were explained in detail, and verbal informed consent was formally obtained via phone calls, in accordance with the Declaration of Helsinki. Furthermore, all data collectors were trained on the importance of maintaining privacy and confidentiality. The study adhered to the principles outlined in the Declaration of Helsinki and relevant national ethical guidelines governing research involving human subjects.

### Study setting, design and participants

The study was conducted in the Silte Zone, Central Ethiopian Region, located at 7.8322° N latitude and 38.2687° E longitude, with an elevation of 1,967 meters above sea level. The zone comprises one government comprehensive specialty hospital, three primary hospitals, and 35 health centers, 14 of which provide ART services. An institution-based retrospective follow-up study was carried out in these public health facilities from January 2017 to December 30, 2022. The source population included all HIV/AIDS patients who initiated ART under the UTT approach, while the study population consisted of randomly selected adult patients from this group.

### Sample size determination and sampling procedure

The sample size for tuberculosis recurrence was calculated using a maximum incidence rate of 50%, due to the absence of prior studies conducted in the country or in regions with comparable socio-demographic and economic conditions.

$$n = (Z\alpha/2)^2 p (1-p)/d^2 = (1.96)^2(.5)(.5)/(.05)^2 \quad n = 384$$

n = final sample size, Z = coefficient of reliability, p = incidence of reoccurrence, d = margin of error

Then, after considering non response rate 15%, final sample size should be n = **442.**

## Sampling procedure

Data collectors used a list of PLHIV patients receiving ART at public health facilities in the Silte Zone to locate the records of study participants who satisfied the inclusion criteria in the patients' follow-up registration book. HIV patients who started ART during the universal test and treat approach between January 1, 2017, and December 30, 2022 were included in the study population using the ART log book as supporting documentation.

Then, we created a sample frame utilizing this data. Patients who satisfied the inclusion criteria were coded and compiled, and a using simple random sampling with computer-generated random number approach was used to select the study subject. And in the way shown in Fig A in S1 Text, the final sample was drawn using a proportional allocation.

## Study variables

**Dependent variables.** Tuberculosis disease

**Independent variable.** Socio demographic (age, sex, residence, catchment area, occupation, educational status, marital status, religion); Clinical (Functional status, BMI, CD4 count, WHO clinical stage, viral load, hemoglobin level, OIs, Non AIDS related chronic disease); Behavioral (ART and anti TB adherence, prophylaxis adherence, substance use); Medication (ART regimen, Cotrimoxazole Preventive Therapy (CPT), Isoniazid Preventive Therapy (IPT), Fluconazole Preventive Therapy (FPT))

## Operational definition

**Censored**: During the follow-up period and at the end of the study, there were no reported cases of tuberculosis infection.
**Event**: diagnosis of TB after initiation of HAART
**Time to incidence of TB:** this is the time from initiation of HAART up to the diagnosis of tuberculosis disease.
**New TB infection:** A TB diagnosis occurring during follow-up in a patient with no prior history of active TB disease at baseline.
**Re-infection of Tuberculosis:** A new episode of TB occurring after completion of previous TB treatment (relapse or reinfection), consistent with national TB guidelines defining relapse categories.
**Recurrence**: Recurrent tuberculosis (TB) refers to a new episode of TB diagnosed after a patient has been declared cured or has completed treatment, with cure defined by WHO as negative sputum smear and culture results in the final month of therapy and on at least one previous occasion [41].
**Loss to follow-up:** defined as an ART patient who has not attended any clinic visit in the past three months and is not recorded as deceased or transferred to another HIV care facility.
**Transfer out:** This occurs when a person living with HIV (PLHIV) receiving care at the selected hospitals transfers their treatment to another health facility.

## ART Adherence

**Good Adherence** (if adherence greater than 95%, meaning the patient missed fewer than 2 doses out of 30 or fewer than 3 doses out of 60, as documented by ART health personnel);

**Fair Adherence** (if adherence between 85% and 94%, corresponding to 3–5 missed doses out of 30 or 3–9 missed doses out of 60, as recorded by ART staff);

**Poor Adherence** (if adherence below 85%, meaning more than 6 missed doses out of 30 or more than 9 missed doses out of 60, according to ART health personnel documentation) [42].

### Functional status

Working: Able to perform usual work in or out of the home;

　Ambulatory: Able to perform daily activities but not work;

　Bedridden: Not able to perform daily activities, in line with WHO functional status categories. [42].

**Substance use:** Self-reported current or historical use of substances including alcohol, khat, and cigarettes, as recorded in medical records [43].

**Catchment area:** The geographic area or population served by the hospital where the patient receives care.

**Anemia:** Defined based on WHO criteria as hemoglobin < 12 g/dL for females and < 13 g/dL for males, as recorded on the medical records [44].

**Isoniazid Preventive Therapy (IPT):** Use of daily isoniazid to prevent active TB disease among people living with HIV who do not have active TB symptoms [22].

**Cotrimoxazole Preventive Therapy (CPT):** Use of cotrimoxazole to prevent opportunistic infections among people living with HIV [22].

**Fluconazole Preventive Therapy (FPT):** Use of fluconazole to prevent fungal opportunistic infections in HIV-positive patients at risk [22].

### Data collection procedure and instrument

Patient medical record numbers were obtained from the chronic care follow-up clinic, and corresponding folders were retrieved from the card room. Data on ART cohorts were extracted using a structured checklist developed from previous literature and HIV care/ART intake and follow-up forms. The extraction tools, prepared in English, were adapted to capture all variables necessary for the study objectives. Data were collected via the Kobo Toolbox platform by healthcare providers from public health facilities in the Silte Zone, under continuous supervision by principal investigators and supervisors, between January 10 and February 10 2023.

### Data analysis

The completeness and consistency of the data were carefully checked before exporting to STATA version 14 for analysis. Descriptive and summary statistics were conducted as appropriate, and results were organized and presented using tables and graphs. The normality of continuous variables was assessed and summarized using either the mean and standard deviation (SD) or the median and interquartile range (IQR), depending on the distribution. Additional coding, categorization, and re-categorization were performed through data transformation for continuous variables and selected categorical variables, respectively.

Survival analysis was conducted since the time from ART initiation to tuberculosis (TB) outcome varied across participants. Patients were censored if they died, transferred out, lost from follow up or did not develop TB disease by the end of follow-up. Kaplan–Meier curves and log-rank tests were used to compare survival across covariate categories, while life tables estimated survival and hazard functions. Bi-variable Cox regression identified candidate predictors ($p < 0.25$) for inclusion in multivariable Cox proportional hazards models. Statistical significance in the multivariable model was set at $p < 0.05$, with associations expressed as adjusted hazard ratios (AHR) and 95% confidence intervals (CI). Model selection relied on log-likelihood and Akaike Information Criterion (AIC), and model adequacy was evaluated using Cox-Snell residuals. The proportional hazards assumption was assessed through graphical diagnostics, goodness-of-fit tests, and time-dependent covariates. Results were presented in tables, figures, and graphical summaries.

# Results

## Baseline characteristics

**Socio-demographic characteristic.** For this study, 442 HIV-positive adult medical records from all Silte Zone ART clinics were examined, with 404 (91.4%) of them being eligible. At the start of ART, the patients median and interquartile range ages were 36 and 15 years, respectively. The sex ratio was about equal. Approximately 60% of the client was an urban resident, and 80% of the client lived in a facility catchment area. In terms of education, roughly one-quarter are either secondary or postsecondary, while the remainder are either primary or have no formal schooling (Table 1).

**Behavioral characteristic.** Of the total number of patients whose medical records were reviewed, 70.3% exhibited good adherence to ART, and more than 80% declared their HIV status. In terms of substance use, around 30% of patients use one or more of khat, alcohol, or cigarettes (Table 2).

**Clinical and immunological characteristics.** When they enrolled in ART, more than 70% of patients were in WHO categories I and II, and more than a quarter were in advanced clinical stages (Stage III and IV). The distribution of CD4 counts among research participants has been shifted to the right, with the median and interquartile range for the patients

**Table 1. Socio-demographic characteristics of patients enrolled on HAART with the UTT approach, 2023.**

| Variables | Category | Frequency | Percentage |
|---|---|---|---|
| Age at initiation ART | <25 | 45 | 11.14 |
| | 25-34 | 118 | 29.21 |
| | 35-44 | 122 | 30.20 |
| | ≥45 | 119 | 29.46 |
| Sex | Male | 206 | 50.99 |
| | Female | 198 | 49.01 |
| Marital status | Never married | 60 | 14.85 |
| | Married | 280 | 69.31 |
| | Separated or Divorced | 24 | 5.94 |
| | Widowed | 40 | 9.9 |
| Educational status | No formal education | 161 | 39.85 |
| | Primary | 147 | 36.39 |
| | Secondary | 46 | 11.39 |
| | Tertiary | 50 | 12.38 |
| Occupation | Governmental employee | 44 | 10.89 |
| | Non-Governmental employee | 22 | 5.45 |
| | Farmer | 65 | 16.09 |
| | Merchant | 129 | 31.93 |
| | Daily laborer | 32 | 7.92 |
| | House wife | 96 | 23.76 |
| | Student | 16 | 3.96 |
| Residence | Urban | 240 | 40.59 |
| | Rural | 164 | 59.41 |
| Catchment area | Reside in facilities catchment area | 322 | 79.7 |
| | Not reside in facilities catchment area | 82 | 20.3 |
| Religion | Muslim | 312 | 77.23 |
| | Orthodox | 58 | 14.36 |
| | Protestant | 32 | 7.94 |

**Table 2. Behavioral, Clinical, Immunological, and Medication-Related Characteristics of Patients Enrolled in HAART under the UTT Approach, 2023.**

| Variables | Categories | Frequency | Percentage |
|---|---|---|---|
| **Adherence** | Good | 248 | 70.30 |
| | Fair | 50 | 12.38 |
| | Poor | 70 | 17.33 |
| **Disclosure status** | Yes | 338 | 83.66 |
| | No | 66 | 16.34 |
| **Substance use** | No | 275 | 68.07 |
| | Yes | 129 | 31.93 |
| **Khat** | No | 248 | 70.30 |
| | Yes | 120 | 29.70 |
| **Alcohol use** | No | 376 | 93.07 |
| | Yes | 28 | 6.93 |
| **Tobacco use** | No | 373 | 92.33 |
| | Yes | 31 | 7.67 |
| **Baseline WHO clinical stag** | Non advanced (Stage 1 &2) | 286 | 70.79 |
| | Advanced (Stage 3 & 4) | 118 | 29.21 |
| **Anemia** | No | 85 | 21.04 |
| | Yes | 319 | 78.96 |
| **CD4 count at initiation of ART** | <200 cells/µl | 87 | 21.53 |
| | 200-499 cells/µl | 243 | 60.14 |
| | ≥ 500 cells/µl | 74 | 18.31 |
| **Hemoglobin level** | Normal | 319 | 78.96 |
| | Below normal | 85 | 21.04 |
| **Functional status at initiation of ART** | Working | 303 | 75.00 |
| | Ambulatory | 74 | 18.32 |
| | Bedridden | 27 | 6.68 |
| **BMI** | >30 | 47 | 11.63 |
| | 18.5-29.9 | 247 | 61.14 |
| | <18.5 | 110 | 27.23 |
| **OIs at initiation of ART** | No | 243 | 57.92 |
| | Yes | 170 | 42.08 |
| **Other non AIDS related comorbidities** | No | 240 | 59.41 |
| | Yes | 164 | 40.59 |
| **ART regimen** | 1e, TDF-3TC-EFV | 397 | 93.81 |
| | 1f, TDF-3TC-NVP | 25 | 6.19 |
| **IPT** | Yes | 282 | 69.8 |
| | No | 122 | 30.2 |
| **CPT** | Yes | 217 | 53.71 |
| | No | 187 | 46.29 |
| **FPT** | Yes | 127 | 31.44 |
| | No | 299 | 68.56 |

being 324 and 220, respectively, with the majority of the individuals having CD4 counts ranging from 200 to 499 cells/l. Subjects' BMI and haemoglobin levels were symmetrically distributed, with mean and standard deviations of 20.43±3.21 and 11.08±1.38, respectively. Approximately 40%, on the other hand, were afflicted with opportunistic infections other than tuberculosis and non-AIDS-related chronic illnesses (Table 2).

**ART and other medication related characteristics.** In this investigation, TDF-3TC-EFV was used in more than 90% of the patients' HAART regimens at the start. More than half (53.71%) of patients had received Cotrimoxazole; roughly 70% were on IPT and 30% were on FPT (Table 2).

### Incidence of new and recurrent tuberculosis

During the four-year retrospective follow-up, 28 patients (6.94%; 5.2% new and 1.73% reinfection) developed tuberculosis, 21 (5.2%) died, 44 (10.89%) were transferred out to other health facilities, 25 (6.19%) were lost to follow-up, and 286 (7.79%) were on follow-up and had not experienced the event until the last visit. Patients were seen for a minimum of 3 months and a maximum of 44 months, for a total of 5,253 person months. The median observation duration was 9 months, with an IQR of 12 months. The overall incidence density rate was 5.33/1000, CI: 3.68, 7.77/1000 person-months of observation (4.01/1000 PM among females and 6.76/1000 PM among males); incidence of new infection was 4/1000 PM (CI: 2.6, 6.13/1000 PM); incidence of reinfection was 1.3/1000 PM (0.63, 2.7/1000 PM). Three-forth (75%) of these occurrences occurred within the first year of monitoring. Survival probability was 0.99, 0.93, 0.90, and 0.87 on the sixth, 12th, and 18th months of the observation, respectively (Table 3).

The overall Kaplan-Meier survival estimate curve showed that incident of tuberculosis was highest in the first year of ART initiation and reduced over the time Incidence of TB was very high among those patients whose CD4 level less than 200 The incidence of Tuberculosis was higher among those patients whose age is > 45 compared to other (Figs 1–3).

To test equality of survival curves of different categorical explanatory variables, Cochran-Mantel Haenszel Log rank test was performed. The test statistics showed that there was a significant difference in survival function for different categorical variables (Table A in S1 Text).

In these study males, study participant who did not disclose their HIV status, underweight and patients with poor ART adherence were had lower survival time as compared to their counterparts and the survival time difference between the groups was found statistically significant (Fig 4).

### Predictors of tuberculosis

Covariates such as gender, age, educational status, client catchment area, disclosure status, functional status, WHO clinical stage, nutritional status, opportunistic infections other than TB, non-HIV/AIDS-related chronic disease CD4 count, IPT, and adherence to ART were found to be eligible for multiple variables regression to identify an independent predictors of tuberculosis infection at a p-value of 0.25 on the bi-variable Cox regression, see Table 4. Marital status, residence, religion, general substance use, khat, alcohol, and tobacco use, haemoglobin level, ART regimen, CPT, and FPT, on the other hand, were determined to be ineligible for tuberculosis prediction (p-value ≥ 0.25) (Table 4).

**Table 3. Life table for tuberculosis survival among HIV patients on HAART with the UTT approach, 2023.**

| Interval in months | Number Entering Interval | Number Withdrawing during Interval | Number Exposed to Risk | Number of Terminal Events | Proportion Terminating | Probability of Surviving | Cumulative Probability Surviving at end of Interval | 95% CI |
|---|---|---|---|---|---|---|---|---|
| 0-6 | 403 | 64 | 371.000 | 5 | 0.01 | 0.99 | 0.99 | 0.97, 0.99 |
| 6-12 | 334 | 131 | 268.500 | 16 | 0.06 | 0.94 | 0.93 | 0.89, 0.95 |
| 12-18 | 187 | 80 | 147.000 | 5 | 0.03 | 0.97 | 0.90 | 0.84, 0.92 |
| 18-24 | 102 | 37 | 83.500 | 2 | 0.02 | 0.98 | 0.87 | 0.81, 0.91 |
| 24-30 | 63 | 26 | 50.000 | 0 | 0.00 | 1.00 | 0.87 | 0.81, 0.91 |
| 30-36 | 37 | 21 | 26.500 | 0 | 0.00 | 1.00 | 0.87 | 0.81, 0.91 |
| 36-42 | 16 | 14 | 9.000 | 0 | 0.00 | 1.00 | 0.87 | 0.81, 0.91 |
| 42-48 | 2 | 2 | 1.000 | 0 | 0.00 | 1.00 | 0.87 | 0.81, 0.91 |

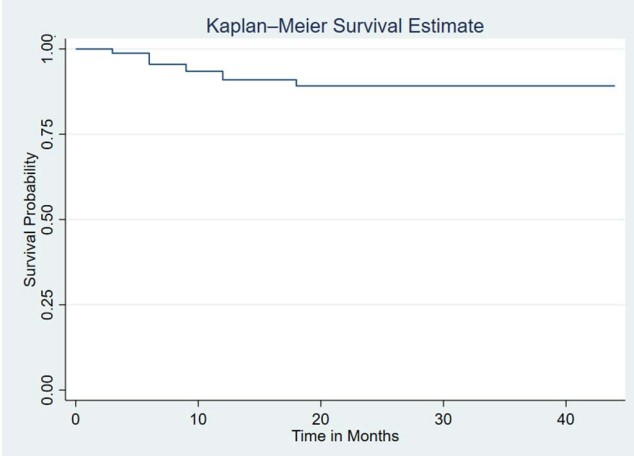

**Fig 1. The overall Kaplan-Meier survival estimate curve of adult HIV patients on HAART with UTT, 2023.**

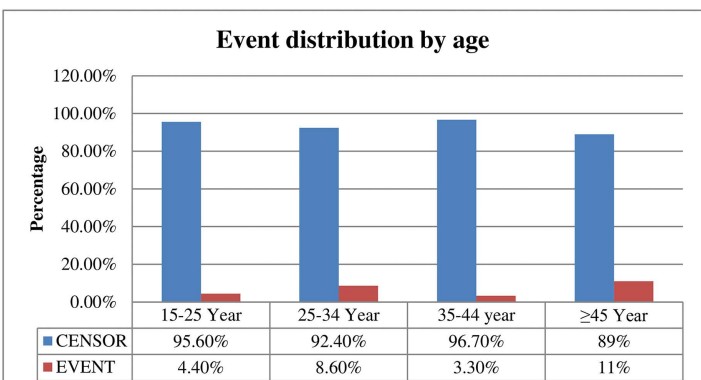

**Fig 2. Tuberculosis distribution by CD4 level among adult HIV patients on HAART with UTT, 2023.**

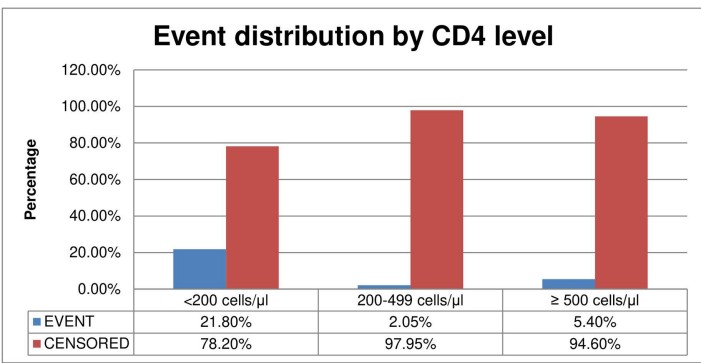

**Fig 3. Tuberculosis distribution by age among adult HIV patients on HAART with the UTT, 2023.**

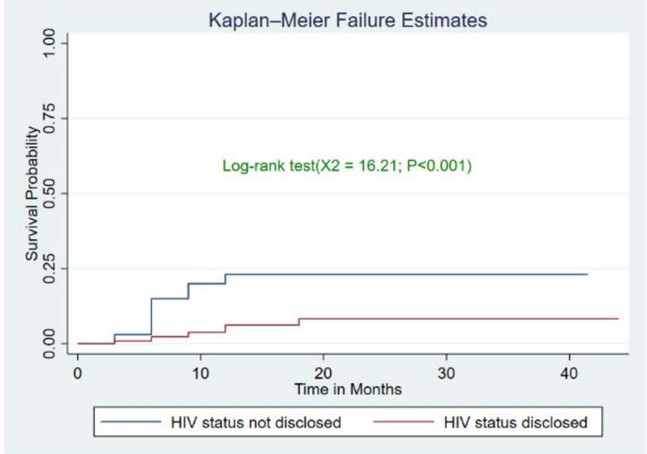
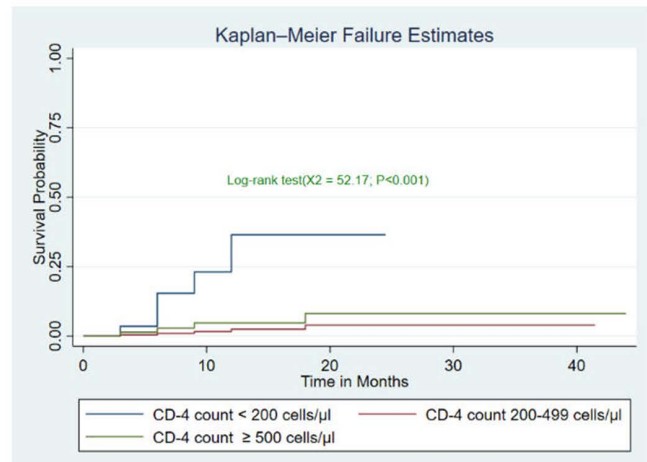
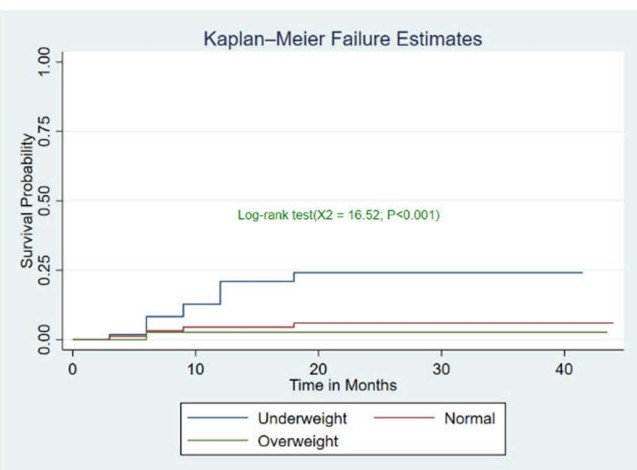
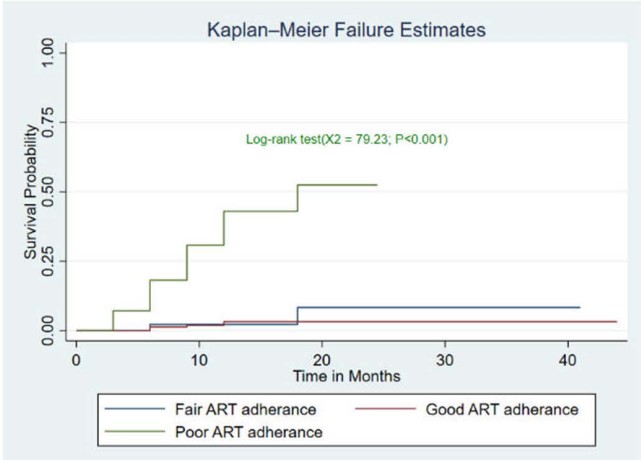

**Fig 4. Kaplan-Meier survival estimates for Sex, Nutritional status, CD4 count and ART adherence among adult HIV patients on HAART with UTT, 2023.**

Sex, disclosure status, nutritional status, CD4 count, and ART drug adherence were revealed as independent predictors of ART-related incidence and recurrence of tuberculosis during multiple-variable Cox regression. Males were five times more likely than females to get tuberculosis at any particular moment [AHR for male: 5.05 (1.79, 14.28)]. In terms of disclosure status, those who did not disclose their HIV status were nearly six times more likely to contract tuberculosis than those who did [AHR for Not disclosed: 6.29 (1.96, 20.19)].

At any given time, those who are underweight have a higher risk of tuberculosis than those who are normal or overweight [AHR for underweight: 3.07 (1.23, 7.65)]. Immunologic findings suggest that the risk of tuberculosis was more than five times higher in individuals with a CD4 count less than 200 compared to those at any given period [AHR for CD4 200 cells/l: 5.63 (1.72, 18.38)]. Furthermore, when compared to their peers, those with poor ART adherence exhibited a seven-fold increase in tuberculosis risk [poor adherence: AHR: 7.05 (1.99-25.01)] (Table 5).

The goodness of fit of the model was assessed by Cox-Snell residuals with the Nelson Aalen cumulative hazard function graph. The graph shows that that the Cox-Snell residuals line is following the Nelson Aalen cumulative hazard function graph which means the model is fit (Fig 5).

**Table 4. Bi-variable Cox regression for predictors of TB among HIV patients enrolled on HAART with UTT approach, 2023.**

| Variables | Category | Survival status | | CHR (95% CI) | P-value |
|---|---|---|---|---|---|
| | | Censored | Event | | |
| **Sex** | Female | 195(94.7) | 11(5.3) | 1 | |
| | Male | 181(91.4) | 17(8.6) | 1.61(0.75, 3.44) | 0.217* |
| **Age** | 15-25 | 43(95.6) | 2(4.4) | 1 | |
| | 25-34 | 109(92.4) | 9(7.6) | 1.68(0.36, 7.77) | 0.507 |
| | 35-44 | 118(96.7) | 4(3.3) | 0.74(0.13, 4.07) | 0.735 |
| | ≥45 | 106(89) | 13(11) | 2.70(0.61, 12.01) | 0.190* |
| **Marital status** | Never married | 55(91.7) | 5(8.3) | 1 | |
| | Married | 262(95.6) | 18(6.4) | 1.28(0.30, 2.10) | 0.806 |
| | Separated /divorced | 23(95.8) | 1(4.2) | 1.65(0.07, 5.18) | 0.647 |
| | Widowed | 36(90) | 4(10) | 2.73(0.44, 6.20) | 0.369 |
| **Education** | No education | 146(90.7) | 15(9.3) | 1 | |
| | Primary | 140(95.2) | 7(4.8) | 0.46(0.18, 1.13) | 0.090* |
| | Secondary | 43(93.5) | 3(6.5) | 0.68(0.19, 2.35) | 0.545 |
| | Tertiary | 47(94) | 3(6) | 0.61(0.17, 2.10) | 0.431 |
| **Occupation** | Government employee | 41(93.2) | 3(6.8) | 0.95(0.26, 3.54) | 0.948 |
| | Farmer | 61(94) | 4(6) | 0.87(0.27, 2.84) | 0.825 |
| | Merchant | 120(93) | 9(7) | 1 | |
| | Daily laborer | 28(87.5) | 4(12.5) | 1.85(0.57 6.03) | 0.303 |
| | House wife | 90(93.8) | 6(6.2) | 0.94(0.33, 2.66) | 0.917 |
| | Student | 15(93.8) | 1(6.2) | 0.85(0.11, 6.76) | 0.883 |
| | Non-government employee | 21(95.5) | 1(4.5) | 0.67(0.08, 5.30) | 0.706 |
| **Residence** | Rural | 155(94.5) | 9(5.5) | 1 | |
| | Urban | 221(92) | 19(8) | 1.57(0.71, 3.46) | 0.267 |
| **Catchment area** | Yes | 304(94.4) | 18(5.6) | 1 | |
| | No | 72(87.8) | 10(12.2) | 2.11(0.97, 4.56) | 0.059* |
| **Adherence** | Good | 278(97.9) | 6(2.1) | 1 | |
| | Fair | 48(96) | 2(4) | 1.87(0.37, 9.28) | 0.442 |
| | Poor | 50(71.5) | 20(28.5) | 18.8(7.45, 47.50) | 0.000* |
| **Disclosure** | Disclosed | 322(95.3) | 16(4.7) | 1 | |
| | Not disclosed | 54(81.8) | 12(18.2) | 4.07(1.92, 8.63) | 0.000* |
| **Substance use** | No | 258(93.8) | 17(6.2) | 1 | |
| | Yes | 118(91.5) | 11(8.5) | 1.43(0.67, 3.06) | 0.350 |
| **Alcohol use** | No | 265(93.3) | 19(6.7) | 1 | |
| | Yes | 111(92.5) | 9(7.5) | 1.79(0.54, 5.95) | 0.338 |
| **Khat use** | No | 351(93.35) | 25(6.65) | 1 | |
| | Yes | 25(89.3) | 3(10.7) | 1.14(0.51, 2.51) | 0.747 |
| **Cigarette** | No | 348(93.3) | 25(6.7) | 1 | |
| | Yes | 28(90.3) | 3(9.7) | 1.85(0.55, 6.20) | 0.315 |
| **Functional status** | Working | 292(96.4) | 11(3.6) | 1 | |
| | Ambulatory | 66(89.2) | 8(10.8) | 3.40(1.36, 8.49) | 0.009* |
| | Bedridden | 18(66.7) | 9(33.3) | 14.14(5.76,34.68) | 0.000* |
| **WHO clinical stage** | I and II | 280(96.8) | 6(3.2) | 1 | |
| | III and IV | 96(81.4) | 22(18.6) | 10.82(4.35,26.88) | 0.000* |

*(Continued)*

**Table 4.** (Continued)

| Variables | Category | Survival status | | CHR (95% CI) | P-value |
|---|---|---|---|---|---|
| | | Censored | Event | | |
| BMI | Underweight | 93(84.6) | 17(15.4) | 3.84(1.76, 8.40) | 0.547 |
| | Normal | 237(96) | 10(4) | 1 | |
| | Overweight | 46(98) | 1(2) | 0.53(0.07, 4.15) | 0.001* |
| OIs other than TB | No | 223(95.3) | 11(4.7) | 1 | |
| | Yes | 153(90) | 17(10) | 2.28(1.07, 4.89) | 0.033* |
| Non-AIDS related chronic disease | No | 228(95) | 12(5) | 1 | |
| | Yes | 148(90.3) | 16(9.7) | 2.12(1.00, 4.49) | 0.049* |
| CD4 count | <200 cells/µl | 68(78.2) | 19(21.8) | 14(5.17, 37.89) | 0.000 |
| | 200-499 cells/µl | 238(97.95) | 5(2.05) | 1 | |
| | ≥ 500 cells/µl | 70(94.6) | 4(5.4) | 2.20(0.59, 8.20) | 0.240 |
| Hemoglobin Level | Normal | 78(91.8) | 7(8.2) | 1 | |
| | Below normal | 298(93.4) | 21(6.6) | 0.88(0.38 2.10) | 0.786 |
| ART regimen | 1e, TDF-3TC-EFV | 352(93) | 27(7) | 1 | |
| | 1f, TDF-3TC-NVP | 24(96) | 1(4) | 0.47(0.06, 3.45) | 0.458 |
| IPT | Yes | 259(92) | 23(8) | 1 | |
| | No | 117(96) | 5(4) | 0.46(0.17, 1.22) | 0.121* |
| CPT | No | 171(91.5) | 16(8.5) | 1 | |
| | Yes | 205(94.5) | 12(5.5) | 0.67(0.31, 1.41) | 0.296 |
| FPT | No | 258(93.2) | 19(6.8) | 1 | |
| | Yes | 118(93) | 9(7) | 1.04(0.473, 2.32) | 0.907 |

## Discussion

A total of 404 patients were retrospectively followed for 5,253 person-months (PM) of observation or 437.75 person-years observation. Patients were followed for a minimum of 3 month and maximum of 44 months and the median observation time was 9 months with IQR 12 months. The overall incidence density rate (IDR) of Tuberculosis in the cohort was 5.33 per-1000-PM (95%CI: 3.68, 7.72/1000), which is equal to 6.39 (95%CI: 4.32, 9.24) per 100PY. Incidence of new infection was 4/1000 PM (CI: 2.6, 6.13/1000 PM); incidence of reinfection was 1.3/1000 PM (0.63, 2.7/1000PM).

The IDR of tuberculosis in this study is consistent with similar study done in Addis Ababa in which the overall IDR among HIV patients enrolled to ART by universal test and treat approach was 4.84 cases(95%CI: 3.83–6.11) per 100 PY over 1529 PY of observation [45]. The current finding is higher than a study conducted at Gurage zone in which IDR of tuberculosis among HIV patient enrolled on UTT was 2.10/100 over 9,766 PM of follow up [46]. This disparity could be attributed to the two research' vastly different follow-up periods. The current study's total PY of observation is nearly one-third that of the Addis Ababa study, and evidence, including this study, indicates that the bulk of occurrences occur in the first year after ART beginning.

However, the incidence in this is lower than other studies conducted in selected ART clinics in Addis Ababa (6.82/100PY during 2140.08 person-years) [47], Debre Markos (6.19/100PY during 1000.22 Person Years follow up) [48] and Afar (8.6/100PY 1377.41 during PY of observation). Study from Tanzania also shows a match more incidence of tuberculosis infection compared to the current study [49].

Regarding the timing of event occurrence, the current finding reveals the risk of tuberculosis infection was higher in the first years of ART initiation (7.33 with 95% CI: 4.99, 10.76/1000 PM) compared to the incidence in the latter years of

**Table 5. Multiple-variable Cox regression for predictors of TB among HIV patients enrolled on HAART with the UTT approach, 2023.**

| Variables | Survival status | | CHR(95%CI) | AHR (95% CI) | P-value |
|---|---|---|---|---|---|
| | Censored | Event | | | |
| **Sex** | | | | | |
| Female | 195(94.7) | 11(5.3) | 1 | 1 | |
| Male | 181(91.4) | 17(8.6) | 1.61(0.75, 3.44) | 5.05 (1.79,14.28) | **0.002** |
| **Age** | | | | | |
| <25 | 43(95.6) | 2(4.4) | 1 | 1 | |
| 25-34 | 109(92.4) | 9(7.6) | 1.68(0.36, 7.77) | 4.93(0.54,44.83) | 0.156 |
| 35-44 | 118(96.7) | 4(3.3) | 0.74(0.13, 4.07) | 1.17(0.91,15.14) | 0.902 |
| ≥45 | 106(89) | 13(11) | 2.70(0.61, 12.01) | 2.24(0.19,26.22) | 0.518 |
| **Client reside within the Catchment Area** | | | | | |
| Yes | 304(94.4) | 18(5.6) | 1 | 1 | |
| No | 72(87.8) | 10(12.2) | 2.11(0.97, 4.56) | 1.88(0.64, 5.55) | 0.247 |
| **Educational Level** | | | | | |
| No education | 146(90.7) | 15(9.3) | 1 | 1 | |
| Primary | 140(95.2) | 7(4.8) | 0.46(0.18, 1.13) | 0.28 (0.08,1.01) | 0.053 |
| Secondary | 43(93.5) | 3(6.5) | 0.68(0.19, 2.35) | 0.22(0.03,1.62) | 0.139 |
| Tertiary | 47(94) | 3(6) | 0.61(0.17, 2.10) | 0.25 (0.04, 1.47) | 0.125 |
| **Disclosure Status** | | | | | |
| Disclosed | 322(95.3) | 16(4.7) | 1 | 1 | |
| Not disclosed | 54(81.8) | 12(18.2) | 4.07(1.92, 8.63) | 6.29(1.96, 20.19) | **0.002** |
| **Functional Status** | | | | | |
| Working | 292(96.4) | 11(3.6) | 1 | 1 | |
| Ambulatory | 66(89.2) | 8(10.8) | 3.40(1.36, 8.49) | 0.77(0.22, 2.69) | 0.692 |
| Bedridden | 18(66.7) | 9(33.3) | 14.14(5.76,34.68) | 0.66(0.18, 2.35) | 0.526 |
| **WHO clinical stage** | | | | | |
| I and II | 280(96.8) | 6(3.2) | 1 | 1 | |
| III and IV | 96(81.4) | 22(18.6) | 10.82(4.35,26.88) | 3.47(0.91,13.20) | 0.068 |
| **Nutritional Status** | | | | | |
| Underweight | 93(84.6) | 17(15.4) | 3.84(1.76, 8.40) | 3.07(1.23, 7.65) | **0.016** |
| Normal | 237(96) | 10(4) | 1 | 1 | |
| Overweight | 46(98) | 1(2) | 0.53(0.07, 4.15) | 0.68(0.07, 6.22) | 0.733 |
| **Opportunistic Infection other than Tuberculosis** | | | | | |
| No | 223(95.3) | 11(4.7) | 1 | 1 | |
| Yes | 153(90) | 17(10) | 2.23(1.07, 4.89) | 0.78(0.29, 2.09) | 0.630 |
| **Non-AIDS related chronic disease** | | | | | |
| No | 228(95) | 12(5) | 1 | 1 | |
| Yes | 148(90.3) | 16(9.7) | 2.12(1.00, 4.49) | 1.08(0.31, 3.81) | 0.899 |
| **CD4 count** | | | | | |
| <200 cells/µl | 68(78.2) | 19(21.8) | 14(5.17, 37.89) | 5.63(1.72, 18.38) | **0.004** |
| 200-499 cells/µl | 238(97.95) | 5(2.05) | 1 | 1 | |
| ≥ 500 cells/µl | 70(94.6) | 4(5.4) | 2.20(0.59, 8.20) | 3.63(0.77, 16.94) | 0.100 |
| **Isoniazid Preventive Therapy** | | | | | |
| Yes | 259(92) | 23(8) | 1 | 1 | |
| No | 117(95.9) | 5(4.1) | 0.46(0.17, 1.22) | 1.05(0.34, 3.19) | 0.931 |
| **ART drug adherence** | | | | | |
| Good | 278(97.9) | 6(2.1) | 1 | 1 | |

*(Continued)*

**Table 5.** (Continued)

| Variables | Survival status | | CHR(95%CI) | AHR (95% CI) | P-value |
|---|---|---|---|---|---|
| | Censored | Event | | | |
| Fair | 48(96) | 2(4) | 1.87(0.37, 9.28) | 1.25(0.19, 9.14) | 0.823 |
| Poor | 50(71.5) | 20(28.5) | 18.8(7.45, 47.50) | 7.05(1.99, 25.01) | **0.002** |

Note: 1 indicates reference group; CI indicates confidence intervals, CHR: Crude Hazard Ratio; AHR: Adjusted Hazard Ratio.

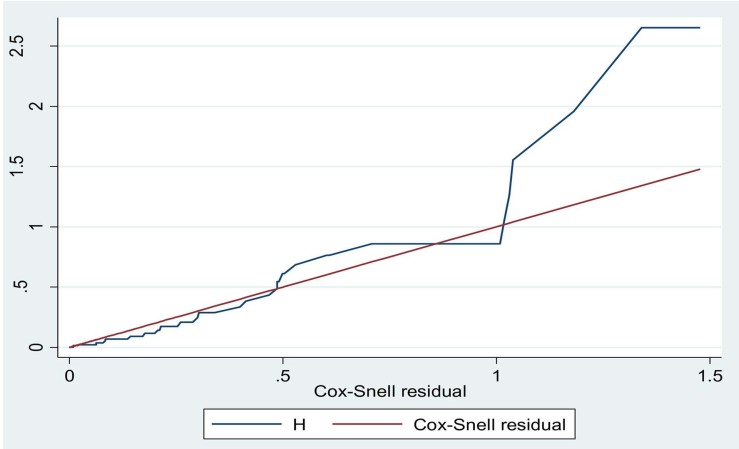

**Fig 5. Cox-Snell residuals with the Nelson Aalen cumulative hazard function graph Goodness of fit of the model.**

ART (1.17 with 95% CI: 0.29, 4.68/1000 PM). Of all the events that happened in our study, three-forth (75%) of the cases occurred in the first year of observation. The cumulative survival probability at the 6th, 12th, 18th, and by the end of the study was 0.99, 0.93, 0.90, and 0.87, respectively. This finding is supported by studies done in Addis Ababa and Debre Markos, in which the highest incidence was observed within the first year of follow-up [47,48].

During multiple variable cox regression the result revealed that sex (being male), disclosure of HIV status (not disclosed HIV status), nutritional status (being underweight), CD4 count (<200 cells) and ART drug adherence (poor ART drug adherence) were statistically significant independent predictors of tuberculosis among HIV patients who initiated ART under the Universal Test and Treat approach.

This study found that, at any given time, the risk of developing tuberculosis among male was 5 times that of their counterpart [AHR: 5.05 (1.79, 14.28)]. This finding is supported by a study done in Tanzania and Ghana; where the risk of Tuberculosis was reduced among female [49,50]. This finding is in line with a study conducted in Nepal in which the prevalence of tuberculosis infection was significantly higher among male PLHIVs than female PLHIVs (AOR: 2.62 (1.176–5.865)) [51]. In addition this finding is also favored by a systematic review and Meta-analysis finding in Sub Saharan Africa, in which being male was a significant factor for an increased incidence of tuberculosis among HIV patients on ART [52].

The higher risk of tuberculosis among male PLHIVs may be due to biological, behavioral, and health-system factors. Men may have weaker immune responses and are more likely to engage in risk behaviors, such as smoking, alcohol use, and high-risk occupational exposure. Delayed health-seeking and poorer adherence to HIV care may further increase their susceptibility.[53].

In terms of disclosure status, the risk of tuberculosis was nearly six times greater among those who did not disclose their HIV status at any given moment [AHR: 6.29 (1.96, 20.19)]. This may be due to patients' fear of disclosing their HIV

status, which may have an indirect negative impact on their adherence to their ART because they are unable to take their medication freely in public, and they may also forget to take the medication and no one reminds them to do so because they did not disclose their HIV status, which is one of the factors contributing to the increased risk of opportunistic infection and failure to suppress viral load [54].

This study indicate that at any given time, those who are underweight have a threefold risk of developing tuberculosis than those who are normal or overweight [AHR: 3.07 (1.23, 7.65)]. This finding is consistent with a study conducted with the same approach in Addis Ababa at St. Peter Hospital and Zewditu Memorial Hospital, which discovered that the risk of tuberculosis is 2.42 times higher among underweight individuals than in normal patients [45].

Studies done under condition, without considering the UTT approach, also have comparable evidence with this study. Such as two studies conducted at Zewditu Memorial Hospital and seven selected ART clinics in Addis Ababa found that the risk of having tuberculosis infection was approximately doubled among underweight patients (AHR = 2.29) [55] and (AHR = 1.91) [47] respectively compared to patients with a normal BMI. Furthermore, the current finding are comparable with a studies conducted in public health institutions in north-east Ethiopia (Afar) [56], north-west Ethiopia (east and west Gojjam) [43], Arba Minch [27] and Sub-Saharan Africa [52].

Underweight HIV patients may have increased catabolic activity, infection, loss of appetite, and decreased intake, which increases the risk of developing TB when compared to persons with a normal BMI. Malnutrition, which reduces immunity and promotes tuberculosis reactivation, could increase susceptibility to tuberculosis in HIV patients who are underweight [57,58].

Immunologic findings suggest that the risk of tuberculosis was more than five times greater in patients with a CD4 count less than 200 at any one time [AHR: 5.63 (1.72, 18.38)]. This finding is consistent with a previous similar study done on HIV patient started ART with UTT approach conducted in Addis Ababa at St. Peter Hospital and Zewditu Memorial Hospital, which found that the risk of tuberculosis infection was three times higher among individuals with a CD4 level of less than 200 cells at any given moment [45]. Evidence from different studies conducted in Ethiopia; Addis Ababa [45], Gondar [28], Debre Markos [34] and Arba Minch [27] found positive evidence for the current conclusion that patients with a CD4 level of 200 cells or less are more likely to get tuberculosis. Furthermore, the findings shown above is confirmed by a meta-regression that found CD4 < 200 cells/mm3 to be a significant positive predictor of TB among HIV patients following the start of ART [59].

This is due to the fact that HIV infection is an immunosuppressive disease that impairs cellular immune responses and raises the risk of opportunistic infections by diverting CD4 + T cells, which is linked to an increased risk of TB development. Low CD4 + cell counts are also linked to recurrence and relapse [60,61].

Moreover, when compared to their counterparts, those with poor ART adherence had a sevenfold greater risk of tuberculosis [AHR: 7.05 (1.99-25.01)]; see Table 5. This finding is confirmed by a study conducted in Addis Ababa at St. Peter Hospital and Zewditu Memorial Hospital, which found that individuals with poor ART adherence were twice as likely to develop tuberculosis [45]. This finding has been verified by a study conducted at Debre Markos referral hospital, which discovered that having fair or poor ART adherence increased the likelihood of contracting tuberculosis [48].

This is due to the fact that treatment adherence is widely viewed as a key element in obtaining optimal outcomes across a wide range of disease states; in the treatment of HIV, poor adherence to treatment has the potential to affect outcomes on numerous levels. Poor adherence to antiretroviral medication (ART) is linked to less efficient viral suppression, which increases the risk of opportunistic infections, most notably tuberculosis [62].

There are a number of limitations to this study. Its retrospective design depended on routinely gathered clinical records, which could be misclassified and had inadequate documentation. Discrepancies with current treatment guidelines may be explained by the fact that ART regimens were recorded at the time of ART initiation and did not capture later regimen switches, such as transitions to tenofovir/lamivudine/dolutegravir during national scale-up. Medical records and self-report were used to document behavioral factors and tuberculosis preventive therapy, which may have resulted in

an overestimation or underestimation. Furthermore, the results may not be entirely applicable to patients starting therapy under more recent national guidelines because many participants started ART during the early stages of Universal Test and Treat implementation, and the observational design prevents causal inference.

## Conclusion

Incidence rate of tuberculosis was reduced among HIV patients who enrolled ART with the Universal Test and Treat approach compared to previous studies. The incidence is high during first year of ART initiation, particularly the first six months. Being male, underweight, not disclosing HIV status, having low CD4 count (<200 cells/µl) and poor ART adherence were an independent predictor for an increased incidence of new or re-infection tuberculosis among HIV patients on ART under the universal test and treat approach.

To reduce tuberculosis risk in HIV patients, early detection through risk-group screening and rapid ART initiation are critical, with close monitoring during the early ART stages when TB prevalence is highest. Patients should be encouraged to disclose their HIV status to close contacts, and ART adherence should be monitored and supported on a frequent basis. Nutritional counseling and weight management should be focused for underweight patients to boost immunity. Furthermore, prospective studies are required to properly quantify the condition and investigate other measurable characteristics using primary patient data.

## Supporting information

**S1 Text. Additional methods and results.**
(DOCX)

**S1 Data. Dataset used for analysis.**
(DTA)

## Author contributions

**Conceptualization:** Abdulbasit Sherfa.

**Data curation:** Abdulbasit Sherfa, Kemal Lemnuro, Mohammed Muze, Abdulmejid Mustefa Shemsu, Musa Jemal, Abas Ali Hassen, Dawit Tafesse Darsema, Belete Birhan, Wolyu Korma.

**Formal analysis:** Abdulbasit Sherfa, Abdulkerim Badegba Isa.

**Funding acquisition:** Abdulbasit Sherfa, Kemal Lemnuro, Mohammed Muze, Wolyu Korma.

**Investigation:** Abdulbasit Sherfa, Kemal Lemnuro, Mohammed Muze, Abdulkerim Badegba Isa, Abdulmejid Mustefa Shemsu, Musa Jemal, Abas Ali Hassen, Dawit Tafesse Darsema, Belete Birhan, Wolyu Korma.

**Methodology:** Abdulbasit Sherfa, Abdulkerim Badegba Isa.

**Project administration:** Abdulbasit Sherfa, Kemal Lemnuro, Mohammed Muze, Abdulmejid Mustefa Shemsu, Musa Jemal, Abas Ali Hassen, Dawit Tafesse Darsema, Belete Birhan, Wolyu Korma.

**Resources:** Abdulbasit Sherfa, Kemal Lemnuro, Mohammed Muze, Wolyu Korma.

**Software:** Abdulbasit Sherfa, Kemal Lemnuro, Abdulkerim Badegba Isa, Abdulmejid Mustefa Shemsu, Musa Jemal, Abas Ali Hassen, Dawit Tafesse Darsema, Belete Birhan, Wolyu Korma.

**Supervision:** Abdulbasit Sherfa, Kemal Lemnuro, Mohammed Muze, Abdulkerim Badegba Isa, Abdulmejid Mustefa Shemsu, Musa Jemal, Abas Ali Hassen, Dawit Tafesse Darsema, Belete Birhan, Wolyu Korma.

**Validation:** Abdulbasit Sherfa, Kemal Lemnuro, Mohammed Muze, Abdulkerim Badegba Isa, Abdulmejid Mustefa Shemsu, Musa Jemal, Abas Ali Hassen, Dawit Tafesse Darsema, Belete Birhan, Wolyu Korma.

**Visualization:** Abdulbasit Sherfa, Kemal Lemnuro, Mohammed Muze, Abdulkerim Badegba Isa, Abdulmejid Mustefa Shemsu, Musa Jemal, Abas Ali Hassen, Dawit Tafesse Darsema, Belete Birhan, Wolyu Korma.

**Writing – original draft:** Abdulbasit Sherfa.

**Writing – review & editing:** Abdulbasit Sherfa, Musa Jemal, Abas Ali Hassen, Dawit Tafesse Darsema, Belete Birhan, Wolyu Korma.

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
