## [Decision Letter · Decision Letter 0]

3 Dec 2025

PGPH-D-25-01412

Time to incidence of tuberculosis and its predictors among adult HIV/AIDS patients who initiated ART by the Universal Taste and Treat approach in Silte Zone, Ethiopia, 2023.

Dear Dr. Sherfa,

Thank you for submitting your manuscript to PLOS Global Public Health. After careful consideration, we feel that it has merit but does not fully meet PLOS Global Public Health’s publication criteria as it currently stands. Therefore, we invite you to submit a revised version of the manuscript that addresses the points raised during the review process.

We look forward to receiving your revised manuscript.

Kind regards,

Damen Haile Mariam, MD, MPH, PhD

Academic Editor

Journal Requirements:

1. Please ensure that your Ethics Statement is available in its entirety at the beginning of your Methods section, under a subheading 'Ethics Statement'.

2. Please upload separate figure files in .tif or .eps format. Also, remove the figures from your manuscript file but keep the legends.

3. We have noticed that you have cited Tables 1 to 8 in the manuscript file but there are no corresponding tables in the manuscript. Please amend your manuscript to include this table, noting that tables should not be uploaded as individual files.

4. We have noticed that you have uploaded Supporting Information files, but you have not included a list of legends. Please add a full list of legends for your Supporting Information files after the references list.

5. In the online submission form, you indicated that “All relevant materials and data are within the manuscript and its supporting files and if any additional data or clarification needed, will be available from the corresponding author upon reasonable request.”.

3. Uploaded as supplementary information.

Additional Editor Comments (if provided):

Reviewer 1:

- General Comments -

- The document requires serious editorial revisions. For instance, the word "taste" is used in the place of "test" in the title of the manuscript.

- The authors have not made distinction between TB disease and TB infection (Refer to the WHO criteria).

- There are also a lot of redundancies.

- Abstract -

- The results should start with describing the cohort characteristics.

- Methods -

- What is the power of the study?

- Results -

- What proportion of patients were given TB preventive therapy? This is part of the care bundle. Most received tenofovir, lamivudine and efavirenz, but nationally it is tenofovir, lamivudine and dolutegravir? Is the reason known?

- Discussion -

- - What are the study limitations?

Reviewer 2:

- Geneal Comments -

- To enhance the manuscript’s contribution and clarity, the reviewer recommends the authors address the following points:

- General Editorial suggestions -

• While the manuscript reads well overall, it would benefit from additional editing and formatting. Specific areas for improvement include:

o Ensure all acronyms are defined at first mention and used consistently thereafter. For example, the “Universal Test and Treat (UTT)” approach is inconsistently referenced with and without the acronym. Other acronyms such as DOT and

SGDs appear without explanation (e.g., on page 3 and page 9).

o Correct typographical errors, such as “aer” instead of “are” and “reveled” instead of “revealed” (page 3, last paragraph).

o Clarify sentence structure. For instance, the sentence ending with “…were considered as common predictors associated with new and re-infection of TB” (page 4, first paragraph) does not logically follow the preceding list of factors.

- Methods -

- The rationale for calculating the sample size based on “tuberculosis recurrence” is unclear. Why was recurrence used instead of the primary outcome of interest—TB incidence?

- The justification for using a 50% maximum incidence rate due to the “absence of prior studies” seems inconsistent with the literature cited in the discussion, including previous meta-analyses. Please clarify.

- Provide clear operational definitions for key study variables, including how TB infection was determined and how independent variables were measured.

- Results -

- On page 8, the manuscript notes the exclusion of 38 patients due to ineligibility. Please specify the eligibility criteria used.

- Table numbering is inconsistent. The results section refers to Tables 1–8, but the attached tables are labeled Tables 9–16. This may be a formatting issue, but the tables should be renumbered accordingly.

- There is a discrepancy in the adherence data: the narrative on page 9 reports 770.3% adherence, which is likely a typographical error. Please verify and correct (likely intended to be 70.3%).

- Avoid using the term “substance abuse” if the data only reflect self-reported use. Additionally, clarify which substances were included beyond alcohol, khat, and cigarettes.

- Consider reducing the number of tables and figures. Presenting only the most critical results in the main text and moving others to supplementary materials may improve readability.

- Discussion -

- Broad claims in the discussion should be supported by references. For example, on page 14, explanations for gender differences in TB incidence (e.g., social behaviors, hormonal differences) are not cited. Similar unsupported statements appear

on pages 15–16.

- On page 15, the statement “Malnutrition, which reduces immunity and promotes tuberculosis activation, could be the source of tuberculosis in HIV patients who are underweight” is misleading. Malnutrition may increase susceptibility but is not a

source of TB. Please revise for accuracy.

- References -

- Review reference 20, which includes conflict of interest disclosures for each author. This may not be necessary in the reference list.

- Check all references for broken links.

- Ensure acronyms are expanded where appropriate. For example, in reference 22, clarify “FMOH” and specify if it refers to the Ethiopian Federal Ministry of Health.

- Tables -

- Renumber tables sequentially from Table 1 to Table 8.

- Include appropriate footnotes for all tables. For instance, Table 15 (likely intended as Table 2) includes asterisks next to p-values, but no explanation is provided. Also, define all acronyms used in the tables.

Reviewers' comments:

Reviewer's Responses to Questions

**Comments to the Author**

1. Does this manuscript meet PLOS Global Public Health’s publication criteria? Is the manuscript technically sound, and do the data support the conclusions? The manuscript must describe methodologically and ethically rigorous research with conclusions that are appropriately drawn based on the data presented.? Is the manuscript technically sound, and do the data support the conclusions? The manuscript must describe methodologically and ethically rigorous research with conclusions that are appropriately drawn based on the data presented.

Reviewer #1: Yes

Reviewer #2: Yes

2. Has the statistical analysis been performed appropriately and rigorously?

Reviewer #1: Yes

Reviewer #2: Yes

3. Have the authors made all data underlying the findings in their manuscript fully available (please refer to the Data Availability Statement at the start of the manuscript PDF file)?

The PLOS Data policy requires authors to make all data underlying the findings described in their manuscript fully available without restriction, with rare exception. The data should be provided as part of the manuscript or its supporting information, or deposited to a public repository. For example, in addition to summary statistics, the data points behind means, medians and variance measures should be available. If there are restrictions on publicly sharing data—e.g. participant privacy or use of data from a third party—those must be specified.requires authors to make all data underlying the findings described in their manuscript fully available without restriction, with rare exception. The data should be provided as part of the manuscript or its supporting information, or deposited to a public repository. For example, in addition to summary statistics, the data points behind means, medians and variance measures should be available. If there are restrictions on publicly sharing data—e.g. participant privacy or use of data from a third party—those must be specified.

Reviewer #1: Yes

Reviewer #2: Yes

4. Is the manuscript presented in an intelligible fashion and written in standard English?

Reviewer #1: Yes

Reviewer #2: Yes

Reviewer #1: The article entitled " Time to incidence of tuberculosis and its predictors among adult HIV/AIDS patients

who initiated ART by the Universal Taste and Treat approach in Silte Zone, Ethiopia" addresses an important public health issue. There are few comments. I see lots of editorial mistakes. One is see above which says "taste". The right word is test. The authors have not made distinction between TB disease and TB infection (Refer to the WHO criteria). There is lot of redundancy; thus it requires diligent editorial work. In the abstract section, the result should start with describing the cohort characteristics. What proportion of patients were given TB preventive therapy? This is part of the care bundle. Most received tenofovir, lamivudine and efavirenz, but nationally it is tenofovir, lamivudine and dolutegravir? Is the reason known? What are the study limitations? What is the power of the study?

Reviewer #2: GENERAL COMMENTS

This is an interesting and well-written manuscript with the potential to make a valuable contribution to scientific literature on the incidence and predictors of tuberculosis (TB) infection among adults living with HIV/AIDS in a developing country context. The authors confirm findings from prior studies that report high levels of incident TB and identify a set of associated risk factors among Ethiopian HIV/AIDS patients. Based on these findings, the authors offer public health and clinical recommendations that, if implemented, could support progress toward TB control targets in Ethiopia and similar settings.

To enhance the manuscript’s contribution and clarity, the reviewer recommends the authors address the following points:

GENERAL EDITORIAL SUGGESTIONS

• While the manuscript reads well overall, it would benefit from additional editing and formatting. Specific areas for improvement include:

o Ensure all acronyms are defined at first mention and used consistently thereafter. For example, the “Universal Test and Treat (UTT)” approach is inconsistently referenced with and without the acronym. Other acronyms such as DOT and SGDs appear without explanation (e.g., on page 3 and page 9).

o Correct typographical errors, such as “aer” instead of “are” and “reveled” instead of “revealed” (page 3, last paragraph).

o Clarify sentence structure. For instance, the sentence ending with “…were considered as common predictors associated with new and re-infection of TB” (page 4, first paragraph) does not logically follow the preceding list of factors.

METHODS

• The rationale for calculating the sample size based on “tuberculosis recurrence” is unclear. Why was recurrence used instead of the primary outcome of interest—TB incidence?

• The justification for using a 50% maximum incidence rate due to the “absence of prior studies” seems inconsistent with the literature cited in the discussion, including previous meta-analyses. Please clarify.

• Provide clear operational definitions for key study variables, including how TB infection was determined and how independent variables were measured.

RESULTS

• On page 8, the manuscript notes the exclusion of 38 patients due to ineligibility. Please specify the eligibility criteria used.

• Table numbering is inconsistent. The results section refers to Tables 1–8, but the attached tables are labeled Tables 9–16. This may be a formatting issue, but the tables should be renumbered accordingly.

• There is a discrepancy in the adherence data: the narrative on page 9 reports 770.3% adherence, which is likely a typographical error. Please verify and correct (likely intended to be 70.3%).

• Avoid using the term “substance abuse” if the data only reflect self-reported use. Additionally, clarify which substances were included beyond alcohol, khat, and cigarettes.

• Consider reducing the number of tables and figures. Presenting only the most critical results in the main text and moving others to supplementary materials may improve readability.

DISCUSSION

• Broad claims in the discussion should be supported by references. For example, on page 14, explanations for gender differences in TB incidence (e.g., social behaviors, hormonal differences) are not cited. Similar unsupported statements appear on pages 15–16.

• On page 15, the statement “Malnutrition, which reduces immunity and promotes tuberculosis activation, could be the source of tuberculosis in HIV patients who are underweight” is misleading. Malnutrition may increase susceptibility but is not a source of TB. Please revise for accuracy.

REFERENCES

• Review reference 20, which includes conflict of interest disclosures for each author. This may not be necessary in the reference list.

• Check all references for broken links.

• Ensure acronyms are expanded where appropriate. For example, in reference 22, clarify “FMOH” and specify if it refers to the Ethiopian Federal Ministry of Health.

TABLES

• Renumber tables sequentially from Table 1 to Table 8.

• Include appropriate footnotes for all tables. For instance, Table 15 (likely intended as Table 2) includes asterisks next to p-values, but no explanation is provided. Also, define all acronyms used in the tables.

**Do you want your identity to be public for this peer review?** For information about this choice, including consent withdrawal, please see our Privacy Policy..

Reviewer #1: **Yes:**Wondwossen Amogne DeguWondwossen Amogne DeguWondwossen Amogne DeguWondwossen Amogne Degu

Reviewer #2: No

---

## [Editor Report · Decision Letter 1]

12 Mar 2026

Time to incidence of tuberculosis and its predictors among adult HIV/AIDS patients who initiated ART by the Universal Test and Treat approach in Silte Zone, Ethiopia, 2023.

PGPH-D-25-01412R1

Dear Mr Sherfa,

We are pleased to inform you that your manuscript 'Time to incidence of tuberculosis and its predictors among adult HIV/AIDS patients who initiated ART by the Universal Test and Treat approach in Silte Zone, Ethiopia, 2023.' has been provisionally accepted for publication in PLOS Global Public Health.

Best regards,

Damen Haile Mariam, MD, MPH, PhD

Academic Editor